# Mycobacterium Tuberculosis Infection after Kidney Transplantation: A Comprehensive Review

**DOI:** 10.3390/pathogens11091041

**Published:** 2022-09-13

**Authors:** Bogdan Marian Sorohan, Gener Ismail, Dorina Tacu, Bogdan Obrișcă, Gina Ciolan, Costin Gîngu, Ioanel Sinescu, Cătălin Baston

**Affiliations:** 1Department of Kidney Transplantation, Fundeni Clinical Institute, 022328 Bucharest, Romania; 2Department of General Medicine, Carol Davila University of Medicine and Pharmacy, 020022 Bucharest, Romania; 3Department of Nephrology, Fundeni Clinical Institute, 022328 Bucharest, Romania; 4Department of Pneumology, Marius Nasta National Institute of Pneumology, 050159 Bucharest, Romania

**Keywords:** mycobacterium, tuberculosis, kidney, transplant, infection, risk, active, latent, prevalence, incidence, graft, rejection, mortality

## Abstract

Tuberculosis (TB) in kidney transplant (KT) recipients is an important opportunistic infection with higher incidence and prevalence than in the general population and is associated with important morbidity and mortality. We performed an extensive literature review of articles published between 1 January 2000 and 15 June 2022 to provide an evidence-based review of epidemiology, pathogenesis, diagnosis, treatment and outcomes of TB in KT recipients. We included all studies which reported epidemiological and/or outcome data regarding active TB in KT, and we approached the diagnostic and treatment challenges according to the current guidelines. Prevalence of active TB in KT recipients ranges between 0.3–15.2%. KT recipients with active TB could have a rejection rate up to 55.6%, a rate of graft loss that varies from 2.2% to 66.6% and a mortality rate up to 60%. Understanding the epidemiological risk, risk factors, transmission modalities, diagnosis and treatment challenges is critical for clinicians in providing an appropriate management for KT with TB. Among diagnostic challenges, which are at the same time associated with delay in management, the following should be considered: atypical clinical presentation, association with co-infections, decreased predictive values of screening tests, diverse radiological aspects and particular diagnostic methods. Regarding treatment challenges in KT recipients with TB, drug interactions, drug toxicities and therapeutical adherence must be considered.

## 1. Introduction

Kidney transplantation (KT) remains the optimal treatment for patients with end-stage renal disease (ESRD) [1]. Nevertheless, infection after KT is still an important limitation for graft and patient outcomes [2,3,4]. One of the most common infections with negative impact post-transplantation is tuberculosis (TB) [5]. It is considered the thirteenth-most common cause of death and the leading infectious cause of death, excluding coronavirus disease 2019 (COVID-19), worldwide [6]. According to World Health Organization (WHO), ~10 million cases of TB (were reported in 2020 worldwide, corresponding to an incidence rate of 127 cases per 100,000 people per year [6,7].

As KT is associated with an immunosuppression status, active TB is higher in KT recipients than in the general population [8,9]. Apart from the immunosuppression condition, there are other several recipient- and donor-associated risk factors that could favor the development of TB in KT [10]. Active TB after KT could arise from reactivation of latent infection in the recipient or donor tissue or can result from de novo infection after transplantation. Endogenous reactivation after KT is the most common form of transmission [11]. In order to limit or prevent the occurrence of active TB post-KT, it is necessary to implement screening measures for both recipients and donors according to the current guidelines [12,13]. 

KT candidates and recipients with TB represent a real challenge regarding the diagnosis and treatment due to atypical or diverse clinical presentation, limitations of screening tests for latent infection, drug interactions and toxicities [14]. The delay in diagnosis and treatment could determine negative consequences, such as graft rejection, graft loss and increased mortality rate [15,16].

This review aimed to provide an evidence-based update regarding epidemiology, risk factors, pathophysiology, type of transmission, diagnostic challenges, treatment challenges and the impact of TB in KT recipients.

## 2. Methods

A literature search on PubMed and Embase electronic databases was performed from 1 January 2000 to 15 June 2022. We used a combination of the following words: “tuberculosis”, “mycobacterium tuberculosis”, “kidney transplant”, “prevalence”, “incidence”, “frequency”, “graft”, “loss”, “failure”, “rejection”, “survival”, “mortality”, “death”, “donor-derived” and “latent”. All studies that provided epidemiological and/or outcomes data regarding TB in KT were included. Articles in languages other than English, articles that evaluated other types of transplantation than kidney only and articles with inadequate information were excluded. All articles were analyzed by two reviewers for inclusion/exclusion criteria, and the process was checked by a third reviewer.

## 3. Epidemiology and Risk Factors for Tuberculosis in Kidney Transplantation

### 3.1. Epidemiology of Tuberculosis in Solid Organ Transplantation

The prevalence of active TB among patients with solid organ transplantation is highly variable according to the geographic area. In areas with low TB endemicity, the prevalence varies between 0.3–6.4%, compared to prevalence of TB in high, endemic areas which could rise to 15.2% [14,17,18]. A recent systematic review and meta-analysis including 60 studies analyzed the prevalence of active TB in solid organ transplant recipients and showed a pooled prevalence of 3% (95% confidence interval (CI): 2–3) [19].

The incidence of active TB in solid organ transplantation is 20–74 times higher than in the general population [12,14]. In a study performed in 16 transplant centers from Spain, which included 4388 solid organ transplantation recipients, the incidence of TB was 512 cases per 100,000 patients per year (95% CI: 317–783), which was 26.6 times higher than in the general population [8]. In another study, conducted in a low TB endemic country on 1989 solid organ transplantation recipients, the incidence of active TB was 41 cases per 100,000 patients per year (95%CI: 15–109), which was 8.5 times higher than the incidence in the general population [20].

### 3.2. Epidemiology of Tuberculosis in Kidney Transplantation

The prevalence of active TB in KT recipients varies from 0.3% to 15.2% (Table 1) [18,21,22,23,24,25,26,27,28,29,30,31,32,33,34,35,36,37,38,39,40,41,42,43,44,45,46,47,48,49,50,51,52,53,54,55,56,57,58,59,60,61,62,63,64] and is higher than in the general population but lower than in patients with lung transplantation [8,9,19,65,66]. Basiri et al. observed a prevalence of 0.3% in a case-control study, which included 12,820 KT recipients from 12 major KT centers in Iran [59]. Additionally, a low prevalence of TB was described by Vandermarliere et al. (0.4%) in a retrospective cohort study from Belgium and by Klote et al. (0.4%) in another study of 15,870 patients from the United States [36,47]. The highest prevalence of active TB in KT recipients (15.2%) was reported by Naqvi et al. in a retrospective cohort study from Pakistan, a country with increased TB endemicity [18]. The pooled prevalence of active TB after KT was analyzed in two systematic reviews and meta-analyses, and the results were relatively similar: 2.51% (95% CI: 2.17–2.85) and 3% (95% CI: 2–3) [19,65]. In another meta-analysis, Al-Efraiji et al. found an unadjusted TB risk ratio of 11.36 (95% CI: 2.97–43.41) times higher in KT recipients, compared to the general population and an adjusted risk ratio for patients on dialysis of 3.62 (95% CI 1.79–7.33) times higher than those from the general population [66].

In a recent meta-analysis, Alemu et al. reported that patients with KT had a pooled incidence of active TB of 2700 (95% CI: 1878–3522) per 100,000 patient-years, which ranged from 340 per 100,000 patient-years in low TB burden countries to 14,680 per 100,000 patient-years in countries with high endemicity [67]. In the same study, the pooled incidence of active TB in KT recipients was higher than in patients with ESRD in pre-dialysis (2700 (95% CI: 1878–3522) vs. 913 (95% CI: 407–1418) per 100,000 person-years) but lower than in those on peritoneal dialysis and hemodialysis (2700 (95% CI: 1878–3522) vs. 3533 (95% CI: 2220–4846) and 5611 (95% CI: 4186–7035) per 100,000 person-years, respectively). Similar to prevalence, some reports showed that active TB incidence in KT recipients is lower than in lung transplant recipients [8,9,11].

### 3.3. Risk Factors for Tuberculosis in Kidney Transplantation

There are several risk factors that predispose KT recipient to develop TB more frequently than the general population [24,26]. The risk is mainly influenced by endemicity of TB in the population, but key factors associated with the recipient, donor and transplantation increase it (Figure 1) [8,10,11,14,24,37,50,64,68,69,70]. Among them, of particular importance are transplant-associated risk factors, such as immunosuppression therapy, presence of acute rejection episodes and chronic graft disfunction [24,32,50,64]. Immunosuppression used in KT impairs T-cell-mediated immunity involved in TB control and favors latent infection reactivation. Some immunosuppressive drugs or combinations certainly increase the risk of TB development: Tcell-depleting agents (anti-thymocyte globulin), cytotoxic T-lymphocyte-associated protein-4 inhibitors (belatacept), calcineurin inhibitors (tacrolimus, cyclosporine), anti-metabolites (mycophenolate, azathioprine) and glucocorticoids [8,9,24,50,64]. Thitisuriyarax et al. showed that acute rejection significantly increases the risk of TB by 7.6 times (95%CI: 1.2–47.9, *p* = 0.03) [50]. Moreover, Basiri et al. showed that the number of rejections after transplant is an independent risk factor for TB appearance [69]. The mechanistic link between rejection and TB development could be indirectly explained by the use of aggressive immunosuppression in the treatment of rejection, leading to TB reactivation. Another transplant-related factor is chronic graft dysfunction, which could increase the risk of TB development by amplifying the immunosuppression status on its own or due to drug overdosing, a condition similar to advanced CKD [68]. A list of recipient-, donor- and transplant-associated risk factors is provided in Figure 1.

**Figure 1 pathogens-11-01041-f001:**
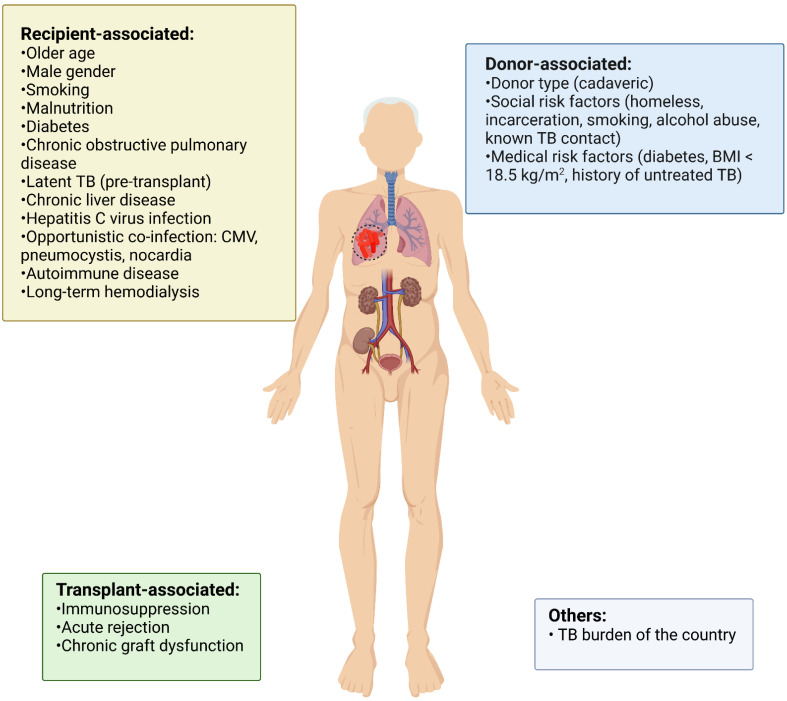
Risk factors for TB development in KT recipients. TB—tuberculosis; CMV—cytoegalovirus; BMI—body mass index; KT—kidney transplantation.

**Table 1 pathogens-11-01041-t001:** Summary of studies evaluating active TB in KT between 2000–2022.

First Author Name (Year)	Population	Recruiting Period	TB Patients/Total Patients	TB Frequency	Time of TB Diagnosis after KT (Months)	TB Type	Graft Loss	Rejection	Mortality
Apaydin et al. (2000) [57]	Turkey	1986–1998	16/274	5.8%	6 (3–119)	Pulmonary (50%) Other (50%)	6.3%	6.3%	31%
Biz et al. (2000) [61]	Brazil	1976–1996	30/1264	2.5%	54 (1.5–216)	Pulmonary (60%) Other (40%)	10%	16.6%	23.3%
Koselj et al. (2000) [62]	Slovenia	1980–1997	8/273	2.9%	34 (1–96)	Pulmonary (62.5%) Other (37.5%)	12.5%	25%	25%
Sharma et al. (2000) [63]	India	NA	21/163	12.9%	42 ± 10	Pulmonary (47.6%) Other (52.4%)	14.3%	NA	23.8%
Vachharajani et al. (2000) [60]	India	1989–1991	16/109	14.7%	6.5 (1-18)	Pulmonary (43.7%) Other (56.3%)	NA	12.5%	18.7%
John et al. (2001) [24]	India	1986–1999	166/1414	13.3%	NA	Pulmonary (48.2%) Other (51.8%)	6%	NA	31.9%
Lezaic et al. (2001) [42]	Yugoslavia	1980–1998	16/456	3.1%	40.5 (1.5–120)	Pulmonary (62.5%) Other (37.5%)	18.7%	0%	37.5%
Naqvi et al. (2001) [18]	Pakistan	1985–2000	130/850	15.2%	12 (10–60)	Pulmonary (52%) Other (48%)	18.5%	22.7%	29%
Melchor et al. (2002) [52]	Mexico	1992–2000	10/545	1.8%	22.5 (2–88)	Pulmonary (60%) Other (40%)	NA	NA	50%
Niewczas et al. (2002) [51]	Poland	1991–2000	15/1289	1.2%	2 (1–44)	Pulmonary (66.6%) Other (33.4%)	26.6%	6.6%	6.6%
Dridi et al. (2003) [53]	Tunisia	1980–2002	5/368	1.3%	27 (3–63)	Pulmonary (60%) Other (40%)	NA	0%	40%
El-Agroudy et al. (2003) [71]	Egypt	1976–1999	45/1200	3.8%	49.8 ± 41.5	Pulmonary (37.7%) Other (62.3%)	35%	55.6%	24.4%
Queipo et al. (2003) [44]	Spain	1980–2000	20/1261	1.6%	20.5 (2–114)	Pulmonary (60%) Other (40%)	5%	NA	15%
Vandermarliere et al. (2003) [47]	Belgium	1963–2001	9/2502	0.4%	64 (5–188)	Pulmonary (67%) Other (33%)	66.6%	33.3%	0%
Klote et al. (2004) [36]	USA	1998–2000	66/15870	0.4%	19.6 ± 12	Pulmonary (62%) Other (38%)	NA	NA	23%
Matuck et al. (2004) [56]	Brazil	1981–2002	44/982	4.5%	36 ± 10.8	Pulmonary (51%) Other (49%)	NA	0%	34.9%
Atasever et al. (2005) [43]	Turkey	1994–2002	20/443	4.5%	53.1 (2–255)	Pulmonary (35%) Other (65%)	NA	NA	28.3%
Chen et al. (2006) [32]	Taiwan	1983–2003	29/756	3.8%	57.9 (1.2–145.2)	Pulmonary (71%) Other (29%)	65.5%	13.6%	41.4%
Ergun et al. (2006) [38]	Turkey	1990–2004	10/283	3.5%	38 (3–81)	Pulmonary (50%) Other (50%)	NA	0%	NA
Ghafari et al. (2007) [55]	Iran	1989–2005	52/1350	3.9%	54.6 (4–140)	Pulmonary (68%) Other (32%)	37%	25%	23%
Kaaroud et al. (2007) [58]	Tunisia	1986–2006	9/359	2.5%	49.6 (3–156)	Pulmonary (55.5%) Other (44.5%)	NA	NA	22.2%
Ram et al. (2007) [23]	India	1989–2005	27/202	13.3%	NA	Pulmonary (33.3%) Other (66.7%)	38.6%	NA	0%
Basiri et al. (2008) [59]	Iran	1964–2003	44/12820	0.3%	25.1 (0.5–78)	Pulmonary (59%) Other (41%)	NA	0%	NA
Chen et al. (2008) [33]	China	1991–2007	43/2333	1.7%	8 (1–156)	Pulmonary (71%) Other (29%)	12.2%	29.3%	21.7%
Ruangkanchanasetr et al. (2008) [49]	Thailand	1987–2007	5/151	3.3%	23 (1–47)	Pulmonary (100%)	NA	20%	0%
Rungruanghiranya et al. (2008) [48]	Thailand	1992–2007	9/270	3.3%	36 (4–115)	Pulmonary (56%) Other (44%)	22.2%	0%	22.2%
Torres et al. (2008) [37]	Spain	1976–2004	16/2012	0.8%	41.9 ± 18.2	Pulmonary (68.7%) Other (22.3%)	NA	37.5%	NA
Guida et al. (2009) [54]	Brazil	1984–2007	23/1342	1.7%	53 ± 49	Pulmonary (43.5%) Other (56.5%)	13%	13%	13%
Canet et al. (2011) [35]	France	1986–2006	74/16,146	0.5%	10 (4–27)	Pulmonary (32.6%) Other (67.4%)	33%	26.5%	6.1%
Ersan et al. (2011) [41]	Turkey	1992–2010	9/320	2.8%	21 (1–150)	Pulmonary (44.4%) Other (55.6%)	22.2%	0%	22.2%
Jung et al. (2012) [27]	South Korea	2000–2010	23/1097	2.1%	26 (3.1–113.2)	NA	NA	0%	14.3%
Ou et al. (2012) [31]	Taiwan	1997–2006	109/4554	2.4%	25.2 (0.1–118.5)	Pulmonary (68.8%) Other (31.2%)	13.8%	NA	22.9%
Boubaker et al. (2013) [40]	Tunisia	1986–2009	16/491	3.2%	23.6 (12.3–190.8)	Pulmonary (50%) Other (50%)	25%	18.7%	12.5%
Marques et al. (2013) [30]	Brazil	2000–2010	43/1549	2.8%	6.5 (0.6–120.8)	Pulmonary (74%) Other (26%)	44%	16%	12%
Rocha et al. (2013) [45] *	Brazil	1998–2010	90/7833	1.1%	80.4 ± 40.8 *	Pulmonary (61.1%) Other (38.9%)	62.5%*	25%*	12.5%*
Higuita et al. (2014) [39]	Colombia	2005–2013	12/641	1.9%	9 (2.3–32.8)	Pulmonary (50%) Other (50%)	8.3%	8.3%	16.6%
Akhtar et al. (2016) [25]	Pakistan	2009–2011	46/910	5%	NA	Pulmonary (48%) Other (52%)	2.2%	NA	13%
Meinerz et al. (2016) [28]	Brazil	2000–2012	60/1737	3.5%	13.4 (1–121.3)	Pulmonary (78.3%) Other (21.7%)	21.6%	1.7%	25%
Costa et al. (2017) [29]	Brazil	1994–2014	34/1604	2.1%	25.5 (1–168)	Pulmonary (47%) Other (53%)	18.5%	44.1%	NA
Gras et al. (2018) [34]	France	2005–2015	32/3974	0.8%	22.3 (8.9–66)	Pulmonary (68%) Other (22%)	15.6%	15.6%	15.6%
Eswarappa et al. (2019) [22]	India	2004–2015	21/244	8.6%	30 (18–54)	Pulmonary (57%) Other (43%)	NA	33.3%	19%
Viana et al. (2019) [64]	Brazil	1998–2014	152/11,453	1.3%	18.8 (7.2–60)	Pulmonary (47.3%) Other (52.7%)	25.6%	17.9%	19%
Park et al. (2021) [26]	South Korea	2011–2015	125/7462	1.7%	NA	NA	8%	NA	16%
Thitisuriyarax et al. (2021) [50]	Thailand	1992–2018	26/787	3.4%	17 (4–59)	Pulmonary (48.1%) Other (51.9%)	37%	11.5%	25.9%
Zou et al. (2021) [21]	China	2005–2020	12/1300	0.9%	22.1(3–120)	Pulmonary (92%) Other (8%)	NA	NA	NA

*—Data available only for abdominal TB; TB—tuberculosis; KT—kidney transplant; NA—not available.

## 4. Transmission and Pathogenesis of Tuberculosis in Kidney Transplant Recipients

According to the natural history of infection, after the aerosol droplets containing mycobacterium tuberculosis (MBT) are inhaled into the lungs, the evolution could be as follows: clearance of MBT by the organism either due to innate immune response or acquired T cell immunity, development of primary disease which means an immediate onset of active disease (first 24 months after primary infection) or latent infection reactivation meaning a late onset of active disease many years following primary infection [12,72]. 

Transmission of TB in KT recipients could be possible according to three scenarios (Figure 2). In the first scenario, active TB could arise after KT as a result of reactivation of the latent infection present in the recipient prior to transplantation (Figure 2A). This form of transmission is the most common found in all solid organ transplantation, including KT [14,73]. In a prospective study conducted by Shu et al., that compared latent infection between KT candidates and recipients, it was observed that the incidence and prevalence of latent infection is higher in KT recipients than candidates [74]. The same study showed that older age, absence of Bacillus Calmette–Guérin vaccine scars, presence of donor specific antibodies and status of KT were factors associated with latent infection [74]. Positive latent infection conversion was found in ~20% of cases within the first 2 years after KT [74]. This finding suggests that a strategy for post-transplantation latent infection evaluation could be helpful in addition to pre-transplant screening. In the second scenario, TB could be transmitted to KT recipients via kidney graft from an infected donor (Figure 2B). This type of transmission is responsible for only 4.8% of cases [75]. In the third scenario, TB could occur as a de novo infection after KT, in a recipient with exposure to a patient with active TB (Figure 2C). This type of transmission is not common; it is associated with very high risk of progression, and it is more frequent in endemic areas [11].

KT generates the favorable rout for TB reactivation in candidates with latent infection. The key point is represented by initiation of induction and maintenance specific immunosuppression [9,24]. The protection against MBT infection is mainly based on cellular immunity, and, more specifically, it depends on T helper 1 (Th1) response [76,77]. Th1-type CD4^+^ T cells and type-1 cytokines are crucial for protection against MBT [77]. Additionally, proliferative capacities of CD4^+^ and CD8^+^ T cells and interferon gamma (IFN-γ) and interleuqin-2 (IL-2) production are essential effectors in the protective response [78]. In patients with latent TB there is an increased signature of CD4^+^ T cells producing IFN-γ and IL-2, compared to active TB patients, in which this type of cells are sparsely represented [79]. Using immunosuppression in the setting of KT could disrupt the protection against TB and increase the risk of reactivation through multiple mechanisms, such as: depletion of all types of T cells, decrease in activation and proliferation of T cells, decrease in IL-2 synthesis, decrease in the production of Th-1 type cytokines or impairment of cellular immunity almost completely [9,24,32,43,80].

## 5. Diagnostic and Treatment Challenges

Diagnosis and treatment of TB can both be challenging due to the particularities associated with KT. These particularities should be known and addressed because the patients’ prognosis depends on the early diagnosis and the appropriate therapeutic approach.

### 5.1. Diagnostic Challenges

#### 5.1.1. Active Tuberculosis

Even if it is known that active TB usually appears in the first year after KT, at a median time of 11.5 months in the case of reactivation after latent infection and earlier in the case of donor-derived infection (in the first 3 months) (Table 2), several distinctive aspects make the diagnosis challenging for clinicians [8,81]. The diagnosis of TB requires a high index of suspicion based on the epidemiological risk, personal history, manifestations and imagistic lesions [11,12]. However, KT recipients have atypical clinical presentations or diverse manifestations, which reduce the clinical suspicion of TB [14]. Furthermore, the probability of association with other co-infections and extrapulmonary localization in ~50% of cases adds a supplementary confusing element to the clinical picture [18,24,25,29,38,39,40,50,54,56,57,60,63,64]. All aforementioned conditions can delay the diagnosis of TB in KT patients [11,12,14,15]. In addition, other challenges in the diagnosis could be linked to paraclinical issues. For example, tuberculin skin test (TST) and interferon-gamma release assay (IGRA) are not useful in the diagnosis of active TB [12]. Additionally, the wide range of radiographic manifestations in pulmonary TB and the frequent need for invasive procedures (bronchoscopy with bronchoalveolar lavage, derange of fluid collections subsequently evaluated by smear and mycobacterium culture and histopathological evaluation) could represent diagnostic challenges [12,13]. Moreover, molecular tests based on rapid nucleic acid amplification techniques, such as mycobacterium tuberculosis complex and resistance to rifampin test (Xpert^®^ MTB/RIF, Cepheid, Sunnyvale, CA, USA), could provide false negative results when mycobacterial load is low [12]. 

Donor-derived TB is of particular interest because even though it is described in a minority of cases, it is associated with severe extrapulmonary manifestations and mortality (Table 2) [73,74]. Nevertheless, there are concerns that the frequency could be higher in areas with increased rates of migration [10]. Donor-derived TB is considered an under-recognized condition with early onset after KT in the majority of cases and should be suspected in KT recipients with one of the following features: non-specific symptoms, frequent fever in the first 3 months after KT, fluid collections, extrapulmonary manifestations or lack of response to empirical antibiotic therapy [81]. Recognizing latent infection or undiagnosed active TB in the kidney donors is critical in preventing post-transplant infection [10]. As active disease in donors is a contraindication for donation, identification of latent TB in deceased donors remains a real challenge in KT, despite the current recommendation for screening [12,13]. Current guidelines (American Society of Transplantation Infectious Diseases Community of Practice and European Society of Clinical Microbiology and Infectious Diseases) recommend a careful evaluation of the epidemiological risk, personal medical history, physical exam and chest radiography in all donors [12,13]. Nevertheless, in deceased donors, the patient’s medical history might be unobtainable, and the screening tests for latent TB (TST and IGRA) have low feasibility and accuracy [10,11,12,70]. In these circumstances, details regarding donor history of previous active TB, specific treatment or exposure to active TB within the last 2 years should be obtained from the donor’s family or relatives [12]. Additionally, if an IGRA test is performed, a series of aspects should be considered—the result might not be available in time, the result could be false negative in donors with head injury due to depressed cell-mediated immunity and, in high-risk donors from low endemic areas with positive tests, the decision of donation should be correlated with personal history and chest imaging [10,12].

#### 5.1.2. Latent Tuberculosis

According to WHO, latent TB infection is defined as a state of persistent immune response to stimulation by MBT antigens with no evidence of clinically manifest active TB [82]. The epidemiology and survey of latent TB after KT remain scarce. Prevalence of latent TB after KT was reported in ~20% of recipients [74,83,84]. Current guidelines provide recommendations for latent TB screening in all KT candidates and donors before transplantation [11,12]. There are no gold standard tests for diagnosing latent TB accurately in KT candidates, but IGRA seems to present some advantages over TST in patients with ESRD [85]. Even so, the evaluation of latent TB in KT recipients is challenging because data regarding prediction capacity of TST and IGRA tests are discordant in this category of patients [86]. Kim et al. showed that IGRA tests have a good predictive potential for latent TB in KT recipient with negative TST [87]. Contrarily, in another study, Hadaya and colleagues observed that IGRA tests had a low sensitivity in KT recipients and cannot be used to exclude latent TB [83]. According to Shu et al., the incidence and prevalence of latent TB in KT recipients is higher than in KT candidates and therefore KT recipients should be more frequently screened [74]. The importance of diagnosis is supported by the fact that undiagnosed and untreated latent TB after KT significantly increases the risk of active TB [74]. 

**Table 2 pathogens-11-01041-t002:** Caser reports of donor-derived TB in KT recipients from 2000 to 2022.

First Author (Year)	Recipient Age (Years)	Recipient Gender	Recipient History of TB	Donor Type	Donor Age	Donor Gender	Donor History of TB	Diagnosis Modality	Diagnosis Time after KT	Symptoms	Localization	Graft Failure	Rejection	Death
Graham et al.(2001) [88]	44	M	NR	DD	69	F	NR	Culture, histopathology	14 months	Anuria	Kidney graft	Yes	No	No
Malone et al. (2007) [89]	53	M	NR	DD	NR	NR	NR	Culture, histopathology	29 months	Nausea	Kidney graft	NR	NR	No
Edathoud et al. (2010) [90]	48	M	NR	DD	26	F	NR	Sputum, BAL, culture, PCR, histopathology	2 months	Fever	Pulmonary, kidney graft, bone marrow	No	No	No
Edathoud et al. (2010) [90]	29	F	NR	DD	26	F	NR	Culture, histopathology	1 month	Fever	Liver	No	No	No
Edathoud et al. (2010) [90]	30	M	NR	DD	NR	NR	NR	Culture, histopathology	21 days	Fever	Bone marrow	No	No	Yes
Al-Nesf et al. (2014) [91]	53	F	NR	LD	NR	NR	NR	Smear, culture, histopathology	2 months	Fever, abdominal pain	Kidney graft	Yes	No	No
Bucher et al. (2016) [92]	69	M	No	DD	67	F	Yes	Culture, PCR	42 days	Skin lesions, cough, anuria	Skin, pulmonary, kidney graft, genitourinary	Yes	No	No
Abad et al. (2018) [81]	30	M	NR	LD	NR	NR	NR	PCR	45 days	NR	Miliary	NR	NR	Yes
Abad et al. (2018) [81]	50	F	NR	DD	46	F	NR	PCR, culture	45 days	Fever	Pulmonary, liver, kidney graft, spleen	NR	NR	Yes
Abad et al. (2018) [81]	23	F	NR	DD	46	F	NR	PCR	47 days	Fever	Miliary, kidney graft	No	No	No
Clemente et al. (2021) [93]	45	M	No	DD	23	F	Yes	Culture	2 months	Fever, nights sweats, chills	Pulmonary, central nervous system, thyroid, kidney graft	Yes	Yes	Yes
Ulisses et al. (2022) [94]	18	F	No	DD	17	M	Unknown	CT, PCR urine, wound	37 days	Fever	Pulmonary, renal, miliary	No	No	No

TB—tuberculosis; KT—kidney transplant; M—male; F—female; LD—Lymphadenopathy; DD—deceased donor; NR—not reported; BAL—bronchoalveolar lavage; CT—computer tomography; PCR—molecular test.

### 5.2. Treatment Challenges

Treatment of KT recipients with TB could be challenging due to drug–drug interactions, drug toxicity and treatment adherence. For this reason, it is recommended that the management of KT patients with TB be carried out by an experienced clinician and special attention must be paid to drug–drug interactions and potentially adverse events [12].

#### 5.2.1. Active Tuberculosis

Although the treatment of active TB in KT recipients respects the principles of treatment for immunocompetent patients, some particularities make it complex and challenging. The treatment of active TB should be promptly started immediately after the diagnosis has been established. Additionally, the epidemiological features from the area of patient’s origin and drug resistance patterns should be assessed [12,13]. 

The optimal period of treatment could vary from 6 to 24 months and, in some cases, based on experts’ opinion, the duration of treatment is recommended to be at least 9–12 months [11,12,13]. American Society of Transplantation Infectious Diseases Community of Practice (AST-IDCOP) guidelines recommends that in case of active uncomplicated pulmonary TB, treatment duration should be at least 6 months, but if cavitary lesions exist or there is a persistent culture-positive sputum after 2 months of therapy, the duration of treatment may be extended to 9 months [11,12,13]. In case of severe disseminated disease or bone and joint disease, treatment duration is recommended for at least 6–9 months [12]. Patients with central nervous system involvement should be treated for at least 9–12 months [12].

According to AST-IDCOP, the first-line treatment should be a four-drug regimen containing rifamycin used both in severe and non-severe cases [12]. Rifamycin is recommended for its sterilization capacity and efficiency but also to reduce the risk of resistance [12]. This standard regimen is similar to that used for the general population and consists of a 2-month intensive phase of isoniazid, rifampicin, pyrazinamide and ethambutol, followed by a 4-month continuation phase of isoniazid and rifampicin [12]. Compared to AST-IDCOP, the European Society of Clinical Microbiology and Infectious Diseases (ESCMID) suggests a standard regimen used for a period longer than 6 months, and, in cases of localized non-severe TB, a regimen without rifampicin could be used if no resistance to isoniazid is present [13]. If a regimen without rifamycin is used, then the 2-month intensive phase should contain isoniazid, ethambutol and pyrazinamide or levofloxacin, followed by a continuation phase of 12–18 months with isoniazid and ethambutol or pyrazinamide. Additionally, if second-line drugs are used, a longer period of treatment is recommended [12,13].

One challenge in the treatment of KT recipients with active TB is the drug interaction between rifampicin and transplant-associated immunosuppression [12,13]. Rifampicin, a potent inducer of cytochrome P450 3A4 and P-glycoprotein, interferes with immunosuppression metabolization [95]. Specifically, rifampicin usage decrease the levels of calcineurin inhibitors (cyclosporine, tacrolimus), the mammalian target of rapamycin (mTOR) inhibitors (sirolimus, everolimus), and affects glucocorticoids metabolization, which increases the risk of rejection [12,13,17,96,97]. Therefore, when a rifampicin-based regimen is used, calcineurin and mTOR inhibitors levels should be closely monitored, the dose of calcineurin and mTOR inhibitor should be increased between three- and five-fold and the glucocorticoid dose should be doubled during treatment and adjusted thereafter to obtain the therapeutic target. Additionally, after the rifampicin is stopped, the immunosuppression doses should be reduced to the value before the start of rifampicin and then adjusted to obtain the therapeutic target [11,12,13]. Despite these recommendations, sometimes the adjustment of immunosuppression is difficult to achieve; the levels of immunosuppressants are suboptimal, and there is still a risk of rejection and graft loss [98,99]. An alternative to rifampicin is rifabutin, which is a weaker inducer of cytochrome P450 3A4 and P-glycoprotein but with similar efficacy [100]. Likewise, even in rifabutin-based regimens, immunosuppression doses could be modified, and levels should be closely monitored [12,13]. Another safe and effective alternative to rifampicin in KT recipients is treatment with fluoroquinolones [101].

Another challenge is linked to the adverse effects of TB therapy, which are more frequent than in the general population. Therefore, some first-line drugs could not be used, and consequently, the treatment period will be increased [11]. Patients treated with anti-TB drugs should be closely monitored for hepatotoxicity (isoniazid, rifampicin, pyrazinamide, ethambutol), neurotoxicity (isoniazid, ethambutol), cytopenia (isoniazid, rifampicin, pyrazinamide, ethambutol), visual disturbances (rifabutin, ethambutol), skin lesions (rifampicin), hyperuricemia (pyrazinamide) or interstitial nephritis (rifampicin, pyrazinamide) [12]. The most common adverse event associated with anti-TB therapy is hepatotoxicity; therefore, liver enzymes should be closely monitored with bi-weekly evaluation during the intensive phase of treatment and monthly thereafter [11].

Treatment adherence could also be an issue in KT recipients. However, implementation of directed observed therapy programs has improved the adherence of patients to anti-TB therapy and their outcomes [12,102].

An additional caveat is that patients with KT could have different degrees of graft function and therefore is very important to evaluate creatinine clearance and adjust the doses for pyrazinamide and ethambutol [103].

Reduction of immunosuppression in the case of severe TB or when a vital organ is involved should be considered. However, there are some concerns regarding the possible occurrence of immune reconstitution inflammatory syndrome, which is associated with the reduction of immunosuppression and the use of rifampicin [11,104,105]. 

#### 5.2.2. Latent Tuberculosis

Treatment of latent TB should be considered only after active TB has been excluded. Treatment of KT recipients with latent TB is important for preventing the risk of reactivation. Treatment in this category of patients is indicated in one of the following conditions: a positive TST or IGRA test, a history of untreated TB, a history of recent contact with an active TB patient and when the kidney graft originates from a donor with known latent TB without chemoprophylaxis, known history of untreated TB or recent exposure to active TB [13]. In the KT setting, the preferred treatment of latent TB is isoniazid 5 mg/kg/day (maximum dose 300 mg/day) for 9 months, supplemented with vitamin B6 [12,13]. A regimen based on rifampicin is not recommended [12,13]. An alternative regimen for KT recipients, mainly for those with high risk, consists of ethambutol and levofloxacin or moxifloxacin [13]. The main adverse event associated to isoniazid is hepatotoxicity, but the risk of liver damage seems to be reduced in KT patients [106]. Nonetheless, evaluation of liver enzymes during treatment, initially bi-weekly for 6 weeks and monthly thereafter, is recommended [12].

## 6. Outcomes

TB in KT is associated with important morbidity and mortality due to immunosuppression status, increased extrapulmonary disease and challenges in diagnosis that delay the initiation of treatment [11].

### 6.1. Rejection

Graft rejection in KT recipients with TB can reach up to 55.6%, often being associated with reduced levels of immunosuppression secondary to calcineurin inhibitors–rifampicin interaction and could be responsible for ~1/3 of graft losses [71]. Vandermarliere et al. observed that 50% of graft failure cases were secondary to acute rejection, and in another study, Guida et al. showed that all kidney graft losses were produced by acute rejection [47,54]. Viana et al showed that treatment of acute rejection before TB significantly increased the risk of graft loss 2.5 times (HR = 2.51, 95%CI: 1.17–5.39, *p* = 0.01) [64].

### 6.2. Graft Loss

The causes of graft loss among KT patients with TB can be directly due to infection, especially in the case of donor-derived TB, or indirectly through the sepsis produced by TBand due to the acute or chronic rejection that occurred after minimization or withdrawal of immunosuppression. Additionally, rejection could be precipitated by suboptimal levels of immunosuppression in the context of rifampicin-based regimen use. The prevalence of graft loss in KT patients with active TB varies from 2.2% to 66.6% (Table 1) [18,23,24,25,26,28,29,30,31,32,33,34,35,39,40,41,42,44,45,46,47,48,50,51,54,55,57,61,62,63,64]. One study that analyzed graft function and survival in KT recipients with active TB showed a prevalence of graft loss of 14.7% and an association of TB with acute kidney injury and incomplete recovery of graft function after treatment [29]. The authors identified that sepsis, acute rejection, interstitial fibrosis and tubular atrophy were causes of graft loss [29]. They also observed that kidney graft function was significantly decreased at the time of diagnosis and during the treatment of TB and remained permanently impaired [29]. According to the univariate analysis, severe TB disease, acute kidney injury stage 2 or 3, acute rejection and value of serum creatinine were risk factors associated with non-recovery of graft function [29]. In another study from Brazil, graft loss frequency in KT recipients with active TB was 25.4% and deceased donor type, treatment of acute rejection within the first year before TB, temporary and definitive discontinuation of immunosuppression were independent risk factors for graft loss [64]. In a study from France with a long follow-up period, graft survival rates in KT patients with TB at 1 year, 5 years and 10 years after KT were 97%, 85% and 67%, respectively [35]. In another long-term follow-up study from Brazil, the 10-year graft survival rate was 56% [30]. The highest rate of graft loss was reported in the study by Vandermarliere et al., in which six out of nine KT patients (66.6%) with active TB lost their grafts [47]. 

### 6.3. Mortality 

In a recent meta-analysis, Mamishi et al. found that mortality rate in solid organ transplantation recipients was 20% [19]. The mortality of patients with TB after KT has been reported to range from 0% to 60% (Table 1) [18,22,23,24,25,26,27,28,30,31,32,33,34,35,36,39,40,41,42,43,44,46,47,48,49,50,51,52,53,55,56,57,58,60,61,62,63,64]. In a retrospective cohort study conducted in 14 KT centers from France, mortality was observed in 6.1% of the KT patients with TB. Out of 74 recipients with TB, 10 developed hemophagocytic syndrome, and among them, mortality was 60% [35]. John et al. showed a high rate of mortality (31.9%) in KT recipients with active TB and found that active TB after 2 years post-KT is an independent risk factor for mortality (HR = 1.84; 95%CI = 1.22–2.78; *p* = 0.003). Noteworthy, 65% of KT recipients with active-TB who died had co-infections with fungi, cytomegalovirus, nocardia, hepatotropic viral infections and chronic liver disease [24]. In a retrospective study from Taiwan conducted by Chen et al., mortality was described in 41.4% of patients with active TB after KT and in 14.4% of cases mortality was associated with anti-TB therapy [32]. An increased rate of mortality (50%) was also reported in a cohort of 545 KT recipients from Mexico. More than half of patients who died received anti-rejection treatment before the TB development and had diabetes, hepatitis C virus infection or fungal infection [52].

## 7. Conclusions

TB in KT is an important opportunistic infection with higher incidence and prevalence than in the general population and is associated with significant negative graft and patient outcomes. Pinpointing the risk factors for both TB development and negative outcomes after KT should be the basis for successful implementation of preventive measures. Additionally, clinicians should recognize the diagnostic and treatment challenges of TB after KT for an optimal management approach. This requires close collaboration between kidney transplant and infectious disease physicians. Donor-derived TB and latent TB in KT are underrecognized conditions that should be carefully evaluated. Development of tests with helpful predictive values, which are not based on T cell immunity, could bring important improvement in the diagnosis of latent TB in KT recipients. Newly discovered regimens or pipeline drugs could have an important contribution in the future limitation of drug–drug interactions, improvement of treatment efficacy and reduction of adverse events. 

## Figures and Tables

**Figure 2 pathogens-11-01041-f002:**
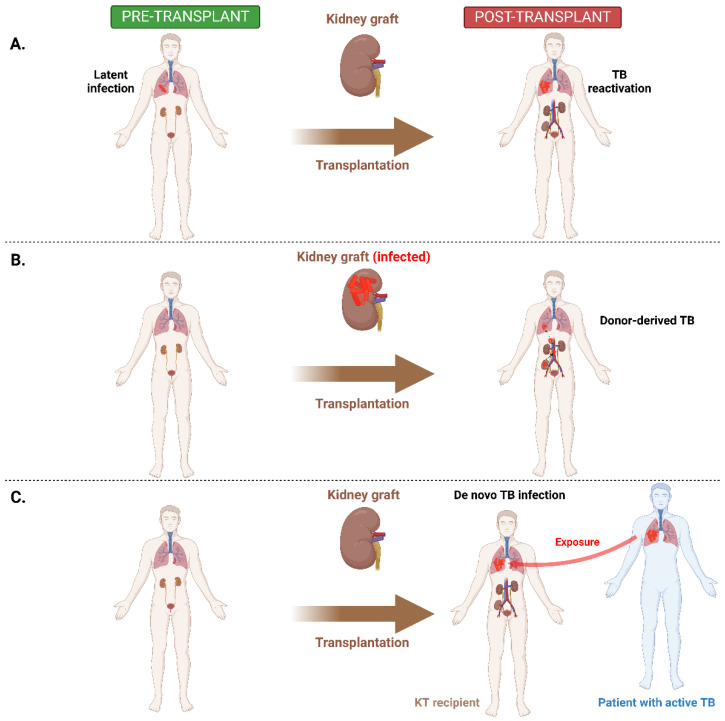
Types of TB transmission in KT recipients. (**A**) Reactivation after KT of a latent infection in recipient. (**B**) Donor-derived infection. (**C**) De novo infection after KT. TB—tuberculosis; KT—kidney transplantation.

## Data Availability

Not applicable.

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
