# Peer review of "Mycobacterium Tuberculosis Infection after Kidney Transplantation: A Comprehensive Review"

_pathogens, 2022, doi:10.3390/pathogens11091041_

Round 1

Reviewer 1 Report

This informative manuscript describes about mycobacterium tuberculosis infection after kidney transplantation including epidemiology, risk factors, transmission, pathogenesis, diagnosis, and impact on graft loss and mortality, and is well written in detail about treatment challenges in cases of mycobacterium tuberculosis infection after kidney transplantation.

Author Response

Thank you for reviewing the manuscript.

Reviewer 2 Report

This paper is a review paper. It is relatively well written based on detailed and extensive data on the topic of tuberculosis in kidney transplatation. However, the English sentences were long and the context was difficult to understand. It seems that the English sentence needs to be corrected for the readers.

example;

Line 17; Tuberculosis (TB) in kidney transplant (KT) recipients in an important----recipients is an important

Line 37, 38

~ 10 million cases of infection with mycobacterium tuberculosis (MBT); not infection but disease -----10 million cases of tuberculosis

Author Response

Dear reviewer,

We would like to thank you for reviewing the manuscript and also for the comments. Please find below the answers and requested changes.

  • Line 17. The mentioned grammar error has been corrected.
  • Lines 37,38. The sentence has been corrected accordingly.
  • Also, we reviewed the entire manuscript and tried to make the sentences more clear where applicable

Reviewer 3 Report

Overall, the whole structure of this study is good and some corrections are recommended for providing clear information. Particularly, I listed the following comments in detail here.
Major concerns:

In abstract, the author needs to mention the ingredients of methods, and materials. Also, the finding of the assay could be added step by step based on material and method. I recommend considering regularly assays and results.

In introduction, there are many spell and grammar mistakes and, also, the sentences are un-regularly disperse. For instance, all of the names and terms should be completely mentioned for the first time in text and so on.  In the different sections, topics pertaining to other sections are sometimes introduced. Additionally, the citations of the literature are not appropriate, and some sentences lack reference.

In methods, the author needs to mention the ingredients of methods, locations, and materials. For example, which methods were used for tests?

In the discussion, discuss your results before relating them to the results of other published work. Also, the author must step by step to come to the results and comparison with others. What is your conclusion? Hence, add a significant statement that must be structured as “what was offered by authors? Do the authors have more thoughts on this field?

In the end, the English language use and grammatical errors require thorough revision by a person familiar with English as a first language.

Author Response

Dear reviewer,

We would like to thank you for reviewing our manuscript, but also for your comments and recommendations. We tried to provide a point-by-point response regarding your concerns.

In abstract, the author needs to mention the ingredients of methods, and materials. Also, the finding of the assay could be added step by step based on material and method. I recommend considering regularly assays and results.

  • We added aspects regarding the methodology in the abstract. Also, we added data that we found in the selected articles, regarding active TB frequency, rejection, graft loss and mortality rates.

In introduction, there are many spell and grammar mistakes and, also, the sentences are unregularly disperse. For instance, all of the names and terms should be completely mentioned for the first time in text and so on.  In the different sections, topics pertaining to other sections are sometimes introduced. Additionally, the citations of the literature are not appropriate, and some sentences lack reference.

  • In the introduction section we checked for grammar and spelling errors and corrected them. Also, we restructured some sentences to provide a continuity of the content. Moreover, we checked the abbreviations without the associated full terms and we modified them accordingly. According to your observation, the missing references were added where necessary.

In methods, the author needs to mention the ingredients of methods, locations, and materials. For example, which methods were used for tests?

  • The present manuscript is not a systematic review and meta-analysis, so the requested details with reference to material and methods do not apply. The manuscript has been written as a comprehensive review without any particularities of a systematic review and meta-analysis and did not imply any analysis test for the accumulated data. We mentioned the methodology aspects of this type of article in section 2.Methods

In the discussion, discuss your results before relating them to the results of other published work. Also, the author must step by step to come to the results and comparison with others. What is your conclusion? Hence, add a significant statement that must be structured as “what was offered by authors? Do the authors have more thoughts on this field?

  • Since the manuscript is not an original paper or a meta-analysis, the discussion section was not addressed. Our article is a comprehensive review and not a systematic review and meta-analysis, so the data were not analyzed using any specific tests. This review was based on a simple description of the data available in the literature at this time for the subjects of interest. Data regarding the frequency of tuberculosis in kidney transplant recipients, the rate of graft loss, rejection and mortality were mentioned in the tables, without any type of heterogeneity or pooling analysis. Also, the manuscript aimed to point-out the types of tuberculosis transmission in kidney transplant recipients, the diagnosis and treatment challenges and the possible solutions based on the current guidelines. Thus, a discussion of our results and comparison with other results are not applicable. In this context, the conclusions were general and covered the given topics.

In the end, the English language use and grammatical errors require thorough revision by a person familiar with English as a first language.

  • English language and grammatical errors have been revised

Reviewer 4 Report

In this manuscript titled “Mycobacterium tuberculosis infection after kidney transplantation: A comprehensive review”, the authors introduced the epidemiology, risk factors, pathogenesis, diagnostic challenges and treatment challenges of TB in patients with kidney transplantation. Overall, this is an interesting review. However, due to insufficient details and references as well as a number of grammar issues, a major revision is needed to improve the quality of this review.

Major issues:

1. The value of diagnosis and treatment of latent TB was not mentioned. Considering the risk associated with treatment of TB in KT patients and graft loss mainly associated with active TB as mentioned in this manuscript, why is it necessary to treat latent TB after KT?

2. There are several grammar issues, please check carefully throughout this manuscript. I pointed out several severe ones below, but other less obvious grammar issues may exist.

3. Line 25-26, please check the grammar.

4. Line 33, I don’t understand this sentence. Specifically, I don’t understand the meaning of “election” here. Please rephrase.

5. Line 53-54, please check the grammar.

6. Line 173-175 and Line 179-182, references are needed.

7. Line 203-204, “the probability of association with other co-infections and extrapulmonary localization in ~ 45% of cases adds a supplementary confusing element to the clinical picture.”, references are needed.

8. Line 226-227, “Current guidelines recommend…”. Please specify which institution or organization generated this guideline.

Author Response

Dear reviewer,

We would like to thank you for the comments and recommendations. Please find below our responses and modifications:

  1. The value of diagnosis and treatment of latent TB was not mentioned. Considering the risk associated with treatment of TB in KT patients and graft loss mainly associated with active TB as mentioned in this manuscript, why is it necessary to treat latent TB after KT?
  • Diagnosis and treatment of latent TB after KT are at least as important as in the case of transplant candidates. Considering that some studies show that the prevalence and incidence of latent TB is higher among KT recipients than KT candidates, the former should be screened more frequently. The diagnosis of latent TB in KT recipients must be followed by treatment whenever possible. Untreated latent TB after KT significantly increases the risk of reactivation and active TB and for this reason it must be treated.
  • Regarding the drug interactions and the risk of graft loss, in the case of latent TB treatment in based only on isoniazid that is not nephrotoxic and do not interact with immunosuppression.
  • We added sentences regarding the importance of diagnosis and treatment of latent TB at (diagnosis and treatment sections for latent TB)

  1. There are several grammar issues, please check carefully throughout this manuscript. I pointed out several severe ones below, but other less obvious grammar issues may exist.
  • Line 25-26- we check the grammar and we modified the sentence
  • Line 33, the word election was changed with optimal
  • Line 53-54- the grammar was checked, and the sentence was modified accordingly
  • Line 173-175 and 179-182- references and sentences were reconsidered and the missing references were added
  • Line 203-204- the percentage was changed to 50% (because 45% was a mistake) and references were added
  • Line 226-227- the name of organizations that generate the guideline has been added accordingly
  • We check carefully throughout the manuscript for other grammar issues and we corrected them when necessary.

Round 2

Reviewer 4 Report

In the revised manuscript, I certainly see the importance of the diagnosis of latent TB, but I still don't see the importance of treating latent TB in these KT patients. The authors argued that latent TB should be treated to reduce the reactivation of TB once latent TB was diagnosed. From this manuscript, it seems that latent TB will not harm patients until they reactivate, then why can't patients be treated after TB reactivation? Treating latent TB certainly brings financial cost to patients and may bring side effects caused by TB drugs too. The authors need to explain the importance and benefits of treating latent TB in more details.

All other concerns resolved.

Author Response

Probably the term “reducing” in this context was not appropriate and thus we changed it with “preventing”. It is absolutely correct that latent TB is not harmful by itself as defined by WHO. Patients with latent TB have no signs of active infections and are diagnosed only by screening tests. The importance of treatment is similar as in KT candidates and guidelines recommend treatment for both categories, candidates and recipients. The importance and benefits are obvious and are based on the fact that treatment of latent infection prevents reactivation of TB, which means ACTIVE INFECTION. The latter is responsible for increased morbidity and mortality as mentioned in the manuscript. For KT recipients the treatment with isoniazid used in latent TB is safe and cost-efficiency issue is for shore favorable than the management of an active TB.

In conclusion we considered that the treatment of latent TB in KT recipients does not require any special explanations other than those mentioned in the manuscript. As the guidelines recommend, it is obvious that any latent infection must be treated to prevent the emergence of active infection, which is really harmful.